published: 09 September 2021

# Urban-Rural Gaps in Breastfeeding Practices: Evidence From Lao People's Democratic Republic

Jordyn T. Wallenborn[1,2]*, Camille B. Valera[3], Sengchanh Kounnavong[4], Somphou Sayasone[4], Peter Odermatt[1,2] and Günther Fink[1,2]

[1]Department of Epidemiology and Public Health, Swiss Tropical and Public Health Institute (Swiss TPH), Basel, Switzerland, [2]University of Basel, Basel, Switzerland, [3]Institute of Global Health, Faculty of Medicine, University of Geneva, Geneva, Switzerland, [4]Lao Tropical and Public Health Institute, Vientiane, Laos

**Objectives:** Breastfeeding rates are decreasing rapidly in many low and middle-income countries, disproportionately affecting urban residences. We use data from Lao People's Democratic Republic to identify primary mechanisms underlying the urban-rural gap in breastfeeding practices.

**Methods:** We used data from the 2017 Lao Social Indicator Survey II. Residence was categorized as large-urban (>1 million), small-urban (<1 Million), and rural. Multivariable logistic regression provided odds ratios and 95% confidence intervals (CI) to identify factors attributing to the urban-rural differences in complying with World Health Organization's breastfeeding recommendations for children <24 months.

**Results:** Mothers in large-urban residences had 3.78 (95% confidence intervals: 1.19, 11.95) and 4.67 (95% CI: 2.30, 9.46) higher odds of non-compliance with exclusive and complementary breastfeeding recommendations, respectively, than mothers living in rural areas in bivariate models. Breastfeeding differentials between small urban and rural residences were largely explained by differences in maternal education and household wealth.

**Conclusion:** Results of our paper suggest large disparities in breastfeeding practices between large-urban, small-urban, and rural residences.

Keywords: rural population, low-and middle-income countries, sociodemographic factors, urban areas, urbanization, breastfeeding, Lao people's democratic republic

**Edited by:**
Carlos Rodriguez-Diaz,
George Washington University,
United States

**Reviewed by:**
Viroj Tangcharoensathien,
Ministry of Public Health, Thailand

***Correspondence:**
Jordyn T. Wallenborn
jordyn.wallenborn@swisstph.ch

**Specialty section:**
This article was submitted to,
a section of the journal International
Journal of Public Health

**Citation:**
Wallenborn JT, Valera CB,
Kounnavong S, Sayasone S,
Odermatt P and Fink G (2021) Urban-
Rural Gaps in Breastfeeding Practices:
Evidence From Lao People's
Democratic Republic.
Int J Public Health 66:1604062.

## INTRODUCTION

Over the past decade, the world's population has steadily become more urbanized—with 56% of the world now living in urban residences [1]. With rapid urbanization in many low- and middle-income countries (LMIC) comes rapid changes to social behavior and health [2], which may contribute to the rapid declines in breastfeeding in many areas [3–6].

A meta-analyses of breastfeeding practices in the 21st century shows that an increase in household income in LMICs is associated with lower rates of continued breastfeeding [3]. Given that economic development in LMIC is usually concentrated in urban areas[7], urbanization may affect breastfeeding in multiple ways. In China, for example, rapid economic growth coincided with an increase in marketing of breastmilk substitutes and significant changes in individual nutritional habits [4].

In the capital of Lao People's Democratic Republic (Lao PDR) — the largest urban area in Lao PDR—less than a quarter (21.0%) of children aged 0–5 months were exclusively breastfed in 2017 [8] compared to 30% in 2012 [9]. This trend continues with complementary breastfeeding, where only 10% of children received breastmilk until age 20–23 months in 2017 compared to 16% in 2012. Nutritional status—a key indicator for adequate complementary feeding—increased in Vientiane. In 2017, 5% of children under five were stunted [8] compared to 3.6% in 2012 [9].

The urban-rural gap in breastfeeding practices has been at the forefront of current debates [10–12]. In our study, we use nationally representative data from data from Lao PDR to: 1) estimate urban-rural differences in compliance with international breastfeeding recommendations, and 2) identify factors that contribute to the urban-rural differences in breastfeeding. We hypothesize, that socioeconomic factors, such as education and wealth, explain the large urban-rural gap in breastfeeding rates. Given that an increasingly large share of the global population lives in large (mega) cities, we separately analyze infant feeding behaviors in large and smaller urban areas.

## METHODS

A cross-sectional study was conducted using data from the 2017 Lao Social Indicator Survey (LSIS) II. LSIS II combines modules of the Multiple Indicator Survey (MICS) and the Demographic and Health Survey (DHS) to maximize survey generalizability and coverage. In collaboration with the Laotian Ministry of Health, the Lao Statistics Bureau and Ministry of Planning and Investment collected data from July to November 2017. LSIS I also combined MICS and the DHS modules and was conducted between 2011–2012 [9]. Data collection for LSIS III is unknown at this time. All women and men aged 15–49 years of age in Lao PDR were eligible. The overarching goal of LSIS II was collection of nationally comparable data on a variety of maternal and child health indicators. Six questionnaires were included in LSIS II. For our study purposes, we focused on the maternal questionnaire. All surveys had over a 99% response rate. More information on study methodology can be found elsewhere [13].

Our main outcome of interest was compliance with the World Health Organization (WHO) breastfeeding recommendations for children under 2 years of age to exclusively breastfeed for 6 months, followed by a combination of breastmilk and complementary foods and liquids until at least 2 years of age (i.e., complementary breastfeeding) [14]. The LSIS II collects detailed breastfeeding information on children under 2 years of age. Using the WHO international breastfeeding recommendations [14], we created a dichotomous (yes; no) variable to display compliance with these recommendations. First, we created a variable measuring compliance with the WHO recommendation to exclusively breastfeed for the first 6 months of life. Exclusive breastfeeding included infants who received breastmilk and no other liquids or supplements besides oral rehydration solution, vitamins, minerals, or other medicines. We also created a variable measuring compliance with WHO

recommendations to complementary breastfed during 6–23 months of age. All children still being breastfed at the time of interview who also received at least one liquid or solid food were included in the complementary breastfeeding variable. All liquid and solid food consumption was self-reported by the mother.

Based on previous literature [6, 15–17] and availability in LSIS II, the following predictors were investigated: marital status, maternal age, highest education attainment of the mother, wealth index, province, residence, attitudes towards domestic violence, prenatal care, skin to skin contact between mother and infant directly after birth, healthcare provider observing breastfeeding within 2 days after birth, healthcare provider counseled on breastfeeding within 2 days of birth, and place of birth. Categorization schemes of all predictors are found in **Table 1**.

Descriptive statistics, including frequencies and column percentages, were calculated to describe population characteristics overall and by residence status (large-urban; small-urban; rural). Chi-square tests were used to identify significant differences in predictors by residence. In order to reduce bias, preserve sample size, and increase statistical power [18], multiple imputation by chained equations (MICE) was used to generate datasets. Multivariable logistic regression provided odds ratios (OR) and 95% confidence intervals (CI) to identify factors associated with breastfeeding practices. Step-wise model building was used to identify factors that helped explain differences in urban-rural breastfeeding practices. The step-wise models included the following groupings: maternal demographics, child factors, socioeconomic status, and healthcare factors. A sensitivity analysis using complete cases was also conducted to substantiate our multiple imputation findings. A $p$-value of 0.05 signifies statistical significance. Data were analyzed using library CRAN, package mice in R [19] and SAS version 9.4 statistical software (SAS, Cary, NC).

## RESULTS

Out of the 11,812 LSIS II participants, we identified 4,654 women with a child less than 24 months for participation in our study. Of these, 175 (4%) lived in a large-urban residence (i.e., Vientiane Capital), 1,099 (24%) in a small-urban residence, and 3,380 (73%) in a rural residence (**Table 1**). Maternal attainment of post-secondary education or more was significantly higher in large-urban residences (36.8%) than small-urban (27.1%) and rural (5.0%) residences. Large, significant disparities were also found in the wealth index, with almost three-quarters (70%) of residents in large-urban settings had the richest wealth index, compared with one-third (33.3%) in small-urban and 4.8% in rural settings. Skin-to-skin contact immediately after birth, breastfeeding counseling within 2 days after birth, and prenatal care were significantly higher in large-urban and small-urban residences. In large-urban settings, 80% of children 0–5 months were not exclusively breastfeed, compared to 51.1% in small-urban settings and 52.3% in rural settings. Similarly for children between 6-23 months, 70.1% were not complementary breastfeed in large-urban settings, 53.1% in small-urban settings, and 35.7% in rural setting (**Table 1**).

**TABLE 1 |** Population characteristics overall and by residence. Lao Social Indicator Survey II, Lao People's Democratic Republic, 2017.

| Characteristic | Study population | Large urban - vientiane capital | Small urban - other provinces | Rural | Chi-square p-value |
|---|---|---|---|---|---|
| | (N = 4,654) | (n = 175) | (n = 1,099) | (n = 3,380) | |
| **Marital Status** | | | | | |
| Married | 4,289 (96.4) | 143 (92.3) | 1,019 (96.9) | 3,125 (96.4) | **0.02** |
| Not Married | 162 (3.6) | 12 (7.7) | 33 (3.1) | 117 (3.6) | — |
| **Maternal Age** | | | | | |
| <20 years | 547 (12.3) | 5 (3.2) | 78 (7.4) | 464 (14.3) | **<0.0001** |
| 20–25 years | 1,625 (36.5) | 28 (18.1) | 323 (30.7) | 1,274 (39.3) | — |
| 26–29 years | 981 (22.1) | 47 (30.3) | 289 (27.5) | 645 (19.9) | — |
| 30–35 years | 870 (19.6) | 48 (31.0) | 256 (24.3) | 566 (17.5) | — |
| >35 years | 426 (9.6) | 27 (17.4) | 106 (10.1) | 293 (9.0) | — |
| **Child Sex** | | | | | |
| Female | 2,243 (48.2) | 88 (50.3) | 558 (50.8) | 1,597 (47.3) | 0.11 |
| Male | 2,411 (51.8) | 87 (49.7) | 541 (49.2) | 1783 (52.8) | — |
| **Child Age At Interview** | | | | | |
| 0–2 Months | 560 (12.3) | 16 (9.9) | 132 (12.2) | 412 (12.4) | 0.11 |
| 3–4 Months | 386 (8.5) | 11 (6.8) | 82 (7.6) | 293 (8.8) | — |
| 5–6 Months | 392 (8.6) | 10 (6.2) | 82 (7.6) | 300 (9.0) | — |
| 7–8 Months | 405 (8.9) | 14 (8.7) | 101 (9.4) | 290 (8.7) | — |
| 9–10 Months | 398 (8.7) | 19 (11.8) | 81 (7.5) | 298 (9.0) | — |
| 11–12 Months | 398 (8.7) | 13 (8.1) | 99 (9.2) | 286 (8.6) | — |
| 13–14 Months | 362 (7.9) | 7 (4.4) | 90 (8.4) | 265 (8.0) | — |
| 15–16 Months | 353 (7.7) | 10 (6.2) | 83 (7.7) | 260 (7.8) | — |
| 17–18 Months | 370 (8.1) | 16 (9.9) | 85 (7.9) | 269 (8.1) | — |
| 19–20 Months | 373 (8.2) | 12 (7.5) | 107 (9.9) | 254 (7.6) | — |
| 21–22 Months | 389 (8.5) | 20 (12.4) | 94 (8.7) | 275 (8.3) | — |
| 23–24 Months | 176 (3.9) | 13 (8.1) | 42 (3.9) | 121 (3.6) | — |
| **Maternal Education** | | | | | |
| No Schooling | 865 (19.5) | 2 (1.3) | 89 (8.5) | 774 (23.9) | **<0.0001** |
| Primary | 1,684 (37.9) | 28 (18.1) | 226 (21.5) | 1,430 (44.2) | — |
| Lower Secondary | 1,007 (22.7) | 37 (23.4) | 273 (26.0) | 697 (21.5) | — |
| Upper Secondary | 386 (8.7) | 31 (20.0) | 178 (16.9) | 177 (5.5) | — |
| Post Secondary or Higher | 503 (11.3) | 57 (36.8) | 285 (27.1) | 161 (5.0) | — |
| **Wealth Index** | | | | | |
| Poorest | 1,257 (27.6) | 1 (0.06) | 66 (6.1) | 1,190 (35.9) | **<0.0001** |
| Second | 1,086 (23.9) | 1 (0.06) | 154 (14.3) | 931 (28.1) | — |
| Middle | 857 (18.8) | 6 (3.7) | 212 (19.7) | 639 (19.3) | — |
| Fourth | 719 (15.8) | 40 (24.8) | 286 (26.6) | 393 (11.9) | — |
| Richest | 630 (13.9) | 113 (70.2) | 359 (33.3) | 158 (4.8) | — |
| **Attitude that domestic violence is not acceptable** | | | | | |
| No | 2,861 (66.1) | 120 (80.0) | 682 (66.2) | 2059 (65.5) | **0.001** |
| Yes | 1,465 (33.9) | 30 (20.0) | 348 (33.8) | 1,087 (34.6) | — |
| **Prenatal Care** | | | | | |
| No | 801 (18.6) | 4 (2.7) | 76 (7.5) | 721 (22.9) | **<0.0001** |
| Yes | 81.4) | 142 (97.3) | 942 (92.5) | 2,427 (77.1) | — |
| **Baby put directly on bare skin of mothers chest after birth** | | | | | |
| No | 2,751 (64.7) | 44 (30.1) | 489 (48.9) | 2,218 (71.5) | **<0.0001** |
| Yes | 1,498 (35.3) | 102 (69.9) | 511 (51.1) | 885 (28.5) | — |
| **Healthcare provider observed child's breastfeeding within 2 days after birth** | | | | | |
| No | 3,774 (89.3) | 95 (67.4) | 842 (85.3) | 2,837 (91.6) | **<0.0001** |
| Yes | 453 (10.7) | 46 (32.6) | 145 (14.7) | 262 (8.5) | — |
| **Healthcare provider counseled on breastfeeding within 2 days after birth** | | | | | |
| No | 3,732 (86.8) | 95 (65.1) | 825 (81.3) | 2,812 (89.6) | **<0.0001** |
| Yes | 566 (13.2) | 51 (34.9) | 190 (18.7) | 325 (10.4) | — |
| **Place of Birth** | | | | | |
| Public Sector | 2,657 (62.2) | 130 (89.7) | 829 (82.2) | 1,698 (54.4) | — |
| Private Medical Sector | 60 (1.4) | 13 (9.0) | 18 (1.8) | 29 (0.9) | — |
| Home | 1,556 (36.4) | 2 (1.4) | 162 (16.1) | 1,392 (44.6) | — |
| **Current exclusive breastfeeding, children 0–6 months** | | | | | |
| No | 588 (47.3) | 24 (80.0) | 136 (51.1) | 496 (52.3) | **0.01** |
| Yes | 656 (52.7) | 6 (20.0) | 130 (48.9) | 452 (47.7) | — |

TABLE 1 | (Continued) Population characteristics overall and by residence. Lao Social Indicator Survey II, Lao People's Democratic Republic, 2017.

| Characteristic | Study population | Large urban - vientiane capital | Small urban - other provinces | Rural | Chi-square p-value |
|---|---|---|---|---|---|
| | (N = 4,654) | (n = 175) | (n = 1,099) | (n = 3,380) | |
| **Current complementary breastfeeding, children 6–23 months** | | | | | |
| No | 1,280 (41.1) | 82 (70.1) | 395 (53.1) | 803 (35.7) | **<0.0001** |
| Yes | 1833 (58.9) | 35 (29.9) | 349 (46.9) | 1,449 (64.3) | — |

*Bold signifies statistical significance.*

**Table 2** displays the association between maternal demographic, child characteristics, socioeconomic status, and other health related factors and non-compliance with WHO's recommendation to exclusively breastfeed during the first 6 months of life. Mothers in Vientiane Capital had 3.5 times the odds of non-compliance with exclusive breastfeeding recommendations in bivariate models (crude OR = 3.78; 95% CI: 1.19, 11.95) and when controlling for maternal demographic factors alone (adjusted OR = 3.56; 95% CI: 1.17, 10.88) compared to mothers residing in rural areas. After controlling for both maternal demographic characteristics and child factors, mothers in large-urban settings had 4.5 times the odds of non-compliance with exclusive breastfeeding recommendations (OR = 4.48; 95% CI: 1.36, 14.73). Once controlling for wealth index and maternal education, estimated associations were attenuated (OR = 2.43; 95% CI: 0.61, 9.64). For small-urban residences in other provinces, the bivariate association and models adjusting for maternal demographics alone or in combination with child factors showed a small increase in the odds of non-compliance with exclusive breastfeeding recommendations (adjusted OR for maternal and child factors = 1.16; 95% CI: 0.84, 1.59). After controlling for the additional socioeconomic variables, mothers who reside in a small-urban residence displayed odds of complying with WHO exclusive breastfeeding recommendations that were similar to mothers in rural areas (adjusted OR: 0.93; 95% CI: 0.61, 1.44). We found similar results from our complete case analysis (**Supplementary Appendix Table S1**).

**Table 3** displays predictors of non-compliance with WHO's recommendations to complementary breastfeed between 6 and 23 months. Mothers in a large-urban residence had 4.7 times the odds of not complying with complementary breastfeeding recommendations in the bivariate model (crude OR: 4.67; 95% CI: 2.30, 9.46). Similar associations were found when controlling for maternal demographics alone (adjusted OR: 4.16; 95% CI: 2.88, 6.02) or in combination with child factors (adjusted OR: 6.64; 95% CI: 3.29, 13.40). However, when controlling for the additional socioeconomic status factors (i.e., wealth index and maternal education), we again found that the association between large-urban residence and non-compliance with WHO's complementary breastfeeding recommendations was attenuated (OR = 2.07; 95% CI: 0.84, 5.08). Mothers in a small-urban residence had 1.9 times the odds of non-compliance with WHO complementary breastfeeding

recommendations in the bivariate model (crude OR: 1.88; 95% CI: 1.52, 2.35). Similar estimates were found when controlling for maternal demographics alone (adjusted OR: 1.78; 95% CI: 1.38, 2.29) or in combination with child factors (adjusted OR: 2.10; 95% CI: 1.45, 3.04). However, once controlling for the additional socioeconomic variables alone (adjusted OR: 1.01; 95% CI: 0.67, 1.51), or in combination with health factors (adjusted OR: 1.04; 95% CI: 0.70, 1.57), the difference between small-urban and rural residences was negligible. Our complete case analysis confirms the wealth relationship and differences in breastfeeding practices by residence (**Supplementary Appendix Table S2**).

## DISCUSSION

Breastfeeding is decreasing rapidly in many urban LMIC settings. In this paper, we analyzed the relationship between residence and breastfeeding practices in Lao PDR as a somewhat representative LMIC facing rapid urbanization and falling breastfeeding rates. Our results highlight the rather pronounced gaps in breastfeeding behaviors between rural and urban areas. On average, children growing up in rural areas are more than twice as likely to be exclusively breastfed in the first 6 months, and also more than twice as likely to benefit from complementary breastfeeding from 6 to 23 months. While most of the breastfeeding gap between small urban and rural areas appears to be explained by differences in maternal education and wealth, the same does not appear to be true for larger urban areas, where substantial breastfeeding gaps are visible even when these factors are adjusted for.

Previous studies conducted in Lao PDR suggest that location of residence[20], encouragement of the child's father [20], television advertisements [20], and formal labor commitments [21] influenced exclusive breastfeeding at 6 months postpartum. Evidence also suggests that ethnic background—which is closely related to geographic region—impact breastfeeding practices [20, 22]. Healthcare workers during antenatal care and in the delivery setting were also shown to have a significant impact on breastfeeding initiation [22]. However, current literature identifying factors influencing breastfeeding in Lao PDR is limited, with only one quantitative study [22] and two qualitative studies [20, 21] investigating these associations.

Over the last 30 years, the Ministry of Health in Lao PDR has attempted to increase breastfeeding rates through standard public health behavior change campaigns, including a Safe Motherhood

**TABLE 2 |** Predictors of non-Compliance with World Health Organization Recommendations to Exclusively Breastfeed during the first 6 months of life, Lao Social Indicator Survey II, Lao People's Democratic Republic, 2017.

| | Bivariate | Model 1 Adjusted for Maternal Demographic Factors | Model 2 Adjusted for Model 1 + Child Factors | Model 3 Adjusted for Model 2 + Socioeconomic Status | Fully adjusted |
|---|---|---|---|---|---|
| **Predictor** | | | Or (95% CI) | | |
| **Residence** | | | | | |
| Large-Urban -Vientiane Capital | 3.78 (1.19, 11.95)* | 3.56 (1.17, 10.88)* | 4.48 (1.36, 14.73)* | 2.43 (0.61, 9.64) | 2.39 (0.61, 9.43) |
| Small- Urban -Other Province | 1.09 (0.81, 1.48) | 1.06 (0.76, 1.47) | 1.16 (0.84, 1.59) | 0.93 (0.61, 1.44) | 0.96 (0.61, 1.50) |
| Rural | References | References | References | References | References |
| **Marital Status** (married vs not married) | 0.64 (0.36, 1.14) | 0.70 (0.39, 1.24) | 0.57 (0.31, 1.07) | 0.63 (0.32, 1.23) | 0.62 (0.31, 1.21) |
| **Maternal Age** | | | | | |
| <20 years | 0.74 (0.43, 1.26) | 0.75 (0.42, 1.33) | 0.69 (0.33, 1.43) | 0.90 (0.42, 1.90) | 0.86 (0.40, 1.83) |
| 20–25 years | 0.67 (0.42, 1.11) | 0.68 (0.41, 1.16) | 0.55 (0.28, 1.10) | 0.64 (0.30, 1.35) | 0.63 (0.29, 1.34) |
| 26–29 years | 0.85 (0.53, 1.37) | 0.84 (0.53, 1.35) | 0.69 (0.39, 1.22) | 0.73 (0.36, 1.48) | 0.73 (0.36, 1.45) |
| 30–35 years | 0.81 (0.47, 1.41) | 0.78 (0.45, 1.35) | 0.71 (0.34, 1.46) | 0.73 (0.38, 1.41) | 0.74 (0.39, 1.40) |
| >35 years | References | References | References | References | References |
| **Child Sex** (Male vs Female) | 0.87 (0.69, 1.09) | — | 0.84 (0.64, 1.11) | 0.84 (0.62, 1.14) | 0.85 (0.62, 1.16) |
| **Children's Age** | | | | | |
| 0–2 months | References | — | References | References | References |
| 3–4 months | 2.65 (1.99, 3.52)*** | — | 2.79 (2.08, 3.75)*** | 2.81 (2.02, 3.93)*** | 2.75 (1.95, 3.89)*** |
| 5–6 months | 12.91 (8.93, 18.65)*** | — | 13.91 (9.52, 20.33)*** | 16.45 (10.85, 24.94)*** | 17.40 (11.53, 26.26)*** |
| **Maternal Education** | | | | | |
| No Schooling | References | — | — | References | References |
| Primary | 0.90 (0.64, 1.27) | — | — | 0.83 (0.53, 1.29) | 0.89 (0.57, 1.40) |
| Lower Secondary | 1.06 (0.74, 1.52) | — | — | 0.75 (0.46, 1.21) | 0.82 (0.50, 1.35) |
| Upper Secondary | 0.97 (0.50, 1.87) | — | — | 0.62 (0.26, 1.48) | 0.69 (0.29, 1.70) |
| Post Secondary or Higher | 1.42 (0.92, 2.17) | — | — | 0.60 (0.30, 1.21) | 0.67 (0.32, 1.40) |
| Wealth Index | | | | | |
| Poorest | References | — | — | References | References |
| Second | 1.20 (0.76, 1.19) | — | — | 1.27 (0.84, 1.94) | 1.37 (0.89, 2.12) |
| Middle | 1.23 (0.85, 1.75) | — | — | 1.51 (0.95, 2.38) | 1.59 (0.99, 2.54) |
| Fourth | 1.66 (1.11, 2.50)* | — | — | 2.37 (1.45, 3.90)** | 2.46 (1.47, 4.12)** |
| Richest | 2.10 (1.30, 3.40)** | — | — | 3.35 (1.49, 7.54)** | 3.50 (1.55, 7.90)** |
| **Attitude that domestic violence is not acceptable** (no vs yes) | 0.90 (0.70, 1.17) | — | — | — | 0.86 (0.59, 1.27) |
| **Prenatal Care** (no vs yes) | 1.04 (0.70, 1.26) | — | — | — | 0.97 (0.63, 1.50) |
| **Baby put directly on bare skin of mothers chest after birth** (no vs yes) | 0.93 (0.72, 1.18) | — | — | — | 1.51 (0.95, 2.32) |
| **Healthcare provider observed child's breastfeeding within 2 days after birth** (no vs yes) | 0.62 (0.33, 0.82)** | — | — | — | 0.54 (0.21, 1.43) |
| **Healthcare provider counseled on breastfeeding within 2 days after birth** (no vs yes) | 0.61 (0.44, 0.86)** | — | — | — | 1.06 (0.51, 2.20) |
| **Place of Birth** | | | | | |
| Public Sector | References | — | — | — | References |
| Private Medical Sector | 1.01 (0.21, 4.94) | — | — | — | 0.51 (0.03, 8.23) |
| Home | 0.75 (0.60, 0.94)* | — | — | — | 0.76 (0.48, 1.21) |

*p-value < 0.05, **p-value < 0.01, ***p-value < 0.001. OR = odds ratio; CI = confidence interval.

program and a large UNICEF supported exclusive breastfeeding promotion campaign; however, these programs were largely unsuccessful [21]. Further, despite the integration of the WHO International Code of Marketing of Breastmilk Substitutes in Lao PDR, 40% of children aged 0–23 months in urban and wealthier households are given commercial breast milk substitutes, compared with 10.6% in rural areas [22].

Marketing of breastmilk substitutes may underline the socioeconomic drivers of poor breastfeeding practices. A qualitative study in Lao PDR found that 75% of mothers report watching television ads promoting infant formula from Thailand, and after seeing these ads, approximately half of mothers wanted to purchase infant formula [21]. Despite the adoption of the WHO International Code of Marketing of

**TABLE 3 |** Predictors of Non-Compliance with World Health Organization Recommendations to Complementary Breastfeed between 6 and 23 months, Lao Social Indicator Survey II, Lao People's Democratic Republic, 2017.

| | Bivariate | Model 1 | Model 2 | Model 3 | Fully adjusted |
|---|---|---|---|---|---|
| | | Adjusted for Maternal Demographic Factors | Adjusted for Model 1 + Child Factors | Adjusted for Model 2 + Socioeconomic Status | |
| **Predictor** | | | Or (95% CI) | | |
| **Residence** | | | | | |
| Large-Urban -Vientiane Capital | 4.67 (2.30, 9.46)*** | 4.16 (2.88, 6.02)*** | 6.64 (3.29, 13.40)*** | 2.07 (0.84, 5.08) | 2.05 (0.78, 5.45) |
| Small- Urban -Other Province | 1.88 (1.52, 2.35)*** | 1.78 (1.38, 2.29)** | 2.10 (1.45, 3.04)** | 1.01 (0.67, 1.51) | 1.04 (0.70, 1.57) |
| Rural | References | References | References | References | References |
| **Marital Status** (married vs not married) | 1.16 (0.75, 1.79) | 1.07 (0.31, 1.47) | 1.25 (0.70, 2.24) | 1.26 (0.69, 2.32) | 1.24 (0.69, 2.22) |
| **Maternal Age** | | | | | |
| <20 years | 0.60 (0.40, 0.90)* | 0.66 (0.43, 1.02) | 0.69 (0.37, 1.29) | 0.78 (0.39, 1.53) | 0.76 (0.37, 1.55) |
| 20–25 years | 0.89 (0.68, 1.16) | 0.85 (0.52, 1.37) | 0.85 (0.45, 1.58) | 0.84 (0.44, 1.59) | 0.81 (0.42, 1.54) |
| 26–29 years | 0.93 (0.64, 1.35) | 0.88 (0.49, 1.56) | 0.91 (0.42, 1.96) | 0.81 (0.36, 1.80) | 0.78 (0.36, 1.69) |
| 30–35 years | 0.88 (0.67, 1.16) | 0.96 (0.54, 1.69) | 0.97 (0.51, 1.82) | 0.84 (0.44, 1.61) | 0.82 (0.42, 1.59) |
| >35 years | References | References | References | References | References |
| **Child Sex** (Female vs Male) | 1.06 (0.92, 1.22) | — | 0.94 (0.76, 1.17) | 0.93 (0.74, 1.17) | 0.94 (0.76, 1.18) |
| **Child Age At Interview** | | | | | |
| 7–8 Months | References | — | References | References | References |
| 9–10 Months | 0.98 (0.59, 1.62) | — | 1.11 (0.65, 1.89) | 1.12 (0.76, 1.66) | 1.16 (0.78, 1.72) |
| 11–12 Months | 1.68 (0.85, 3.31) | — | 1.80 (1.14, 2.86)* | 2.00 (1.18, 3.38)* | 2.06 (1.17, 3.61)* |
| 13–14 Months | 2.93 (1.78, 4.82)*** | — | 2.72 (1.84, 4.01)*** | 3.17 (2.00, 5.03)** | 3.20 (2.00, 5.13)** |
| 15–16 Months | 5.98 (3.41, 10.49)*** | — | 6.83 (5.06, 9.22)*** | 8.93 (6.42, 12.44)*** | 9.19 (6.75, 12.51)*** |
| 17–18 Months | 10.88 (7.40, 15.99)*** | — | 11.82 (8.88, 15.74)*** | 14.15 (9.19, 21.79)*** | 15.15 (9.31, 24.65)*** |
| 19–20 Months | 12.23 (7.86, 19.02)*** | — | 14.82 (10.51, 20.91)*** | 19.48 (11.23, 33.77)*** | 20.37 (11.13, 37.27)*** |
| 21–22 Months | 16.56 (8.62, 31.83)*** | — | 16.99 (12.82, 22.51)*** | 21.68 (15.34, 30.64)*** | 22.48 (15.67, 32.28)*** |
| 23–24 Months | 26.41 (15.91, 43.83)*** | — | 31.66 (16.87, 59.42)*** | 47.74 (27.94, 81.55)*** | 49.99 (27.79, 89.91)*** |
| **Maternal Education** | | | | | |
| No Schooling | References | — | — | References | References |
| Primary | 1.96 (1.70, 2.25)** | — | — | 1.68 (1.31, 2.16)** | 1.58 (1.27, 1.96)** |
| Lower Secondary | 2.44 (1.98, 3.00)** | — | — | 1.69 (1.22, 2.34)** | 1.56 (1.22, 1.98)** |
| Upper Secondary | 3.16 (2.09, 4.78)*** | — | — | 1.60 (0.92, 2.78) | 1.51 (0.86, 2.67) |
| Post Secondary or Higher | 5.01 (3.03, 8.28)*** | — | — | 3.24 (1.60, 6.55)** | 3.03 (1.46, 6.30)* |
| **Wealth Index** | | | | | |
| Poorest | References | — | — | References | References |
| Second | 1.37 (1.10, 1.69)** | — | — | 1.35 (1.05, 1.73)* | 1.26 (0.99, 1.60) |
| Middle | 2.48 (1.62, 3.79)** | — | — | 2.76 (1.49, 5.09)** | 2.60 (1.36, 4.97)* |
| Fourth | 3.54 (2.82, 4.46)*** | — | — | 3.90 (2.38, 6.39)*** | 3.72 (2.01, 6.90)** |
| Richest | 4.74 (3.77, 5.96)*** | — | — | 4.30 (3.11, 5.93)*** | 4.10 (2.63, 6.38)*** |
| **Attitude that domestic violence is not acceptable** (no vs yes) | 1.12 (1.01, 1.24)* | — | — | — | 1.02 (0.90, 1.17) |
| **Prenatal Care** (no vs yes) | 0.50 (0.32, 0.79)* | — | — | — | 0.64 (0.34, 1.22) |
| **Baby put directly on bare skin of mothers chest after birth** (no vs yes) | 0.82 (0.59, 1.13) | — | — | — | 1.24 (0.82, 1.86) |
| **Healthcare provider observed child's breastfeeding within 2 days after birth** (no vs yes) | 0.63 (0.43, 0.92)* | — | — | — | 0.85 (0.56, 1.28) |
| **Healthcare provider counseled on breastfeeding within 2 days after birth** (no vs yes) | 0.64 (0.41, 1.01) | — | — | — | 1.19 (0.67, 2.13) |
| **Place of Birth** | | | | | |
| Public Sector | References | — | — | — | References |
| Private Medical Sector | 2.47 (1.14, 5.35)** | — | — | — | 2.20 (0.65, 7.42) |
| Home | 0.58 (0.42, 0.78)** | — | — | — | 1.08 (0.83, 1.41) |

*p-value < 0.05, **p-value < 0.01, ***p-value < 0.001. OR = odds ratio; CI = confidence interval.

Breastmilk Substitutes in Lao PDR, a lack of regulation may undermine the codes effectiveness. Further, in emerging economies where families have disposable income, promotion of other breastmilk substitutes, such as baby food, have also been heavily marketed [23].

We also found that wealth index is a strong, and consistent predictor of compliance with WHO's breastfeeding recommendations. Our results support a recent brief from WHO which states that more affluent women in LMIC's breastfeed a shorter duration than poorer women [24]. Our results also suggest that the large differences in breastfeeding practices between women residing in urban vs rural areas can be partially attributed to socioeconomic status. Wealth index is a potential proxy for employment status, with wealthier women working more outside of the household. Participation in the female-workforce is a well-known and strong predictor of suboptimal breastfeeding practices[22, 25] — which could help explain our findings.

## Limitations

To our knowledge, our study is the second quantitative analysis of breastfeeding practices in Lao PDR—strengthening our understanding of the urban-rural gap in breastfeeding. The large and extensive dataset from LSIS II allows investigation of important confounding factors, such as wealth and breastfeeding promotion strategies used in breastfeeding interventions (i.e., early skin-to-skin contact). Unlike other breastfeeding research, LSIS II breastfeeding measurements does not suffer from recall bias since the information is cross-sectional (e.g., are the participants breastfeeding at time of interview?); however, temporality between factors cannot be established and causality is not determined. Similarly, residual confounding from maternal-infant bonding and employment status could not be investigated. Our study is also limited by the relatively small sample size from large urban areas. We hope that the substantial intra-urban differentials in breastfeeding behavior can be investigated in future research.

## Conclusion

Results of our paper suggest large disparities in breastfeeding practices between large-urban, small-urban, and rural residences. Increasing education, rising household incomes as well as the trend towards large cities will likely result in rapidly declining breastfeeding rates over the next decade at the global level unless governments identify policy measures that counteract this trend.

Additional research is needed in Lao PDR as well as in LMICs more generally to understand the mechanism behind rapidly declining breastfeeding rates in urban settings, and the role of socioeconomic status.

## ETHICS STATEMENT

Ethical review and approval was not required for the study on human participants in accordance with the local legislation and institutional requirements. The patients/participants provided their written informed consent to participate in this study.

## AUTHOR CONTRIBUTIONS

JW conceptualized and designed the study, carried out the initial analyses, drafted the initial manuscript, and reviewed and revised the manuscript. GF conceptualized and designed the study, and reviewed and revised the manuscript. PT, CV, SK, and SS reviewed the manuscript for important intellectual content, and reviewed and revised the manuscript. All authors approved the final manuscript as submitted and agree to be accountable for all aspects of the work.

## FUNDING

We thank the Publication Fund of the University of Basel for Open Access for covering the open access publication fees.

## CONFLICT OF INTEREST

The authors declare that the research was conducted in the absence of any commercial or financial relationships that could be construed as a potential conflict of interest.

## SUPPLEMENTARY MATERIAL

The Supplementary Material for this article can be found online at: https://www.ssph-journal.org/articles/10.3389/ijph.2021.1604062/full#supplementary-material

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
