## [Reviewer comments · International Journal of Public Health]

Peer Review Report

Review Report on Urban-Rural Gaps in Breastfeeding Practices: Evidence from Lao People's Democratic Republic

Original Article, Int J Public Health

Reviewer: Viroj Tangcharoensathien

Submitted on: 17 May 2021

Article DOI: 10.3389/ijph.2021.1604062

EVALUATION

Q 1 Please provide your detailed review report to the authors. The editors prefer to receive your review structured in major and minor comments. Please consider in your review the methods (statistical methods valid and correctly applied (e.g. sample size, choice of test), is the study replicable based on the method description?), results, data interpretation and references. If there are any objective errors, or if the conclusions are not supported, you should detail your concerns.

Overall comments

- This is an interesting paper, providing evidence in Lao PDR on breast feeding practice comparing three areas: large urban, small urban and rural settings through nationally representative survey (Social Indicator/MICS survey). The paper is clearly written and easy to read.
- The analysis, statistical methods are clearly described, and findings are interesting. It depicts gap of EBF and continue breast feeding and complementary feeding until 24 months comparing three settings.
- This paper maximizes the use the national dataset to document evidence for policy decision. Findings are not new and not unexpected; it confirms international evidence on declining EBF rate in urban settings.
- Limitation is clearly identified, due to the nature of questionnaire, it is not possible to document the influence of aggressive market promotion by BMS industries, except review from literatures that pregnant women have seen televised marketing from Thailand which can influence their decision to use BMS.

Specific comments

1. The introduction is rather short. To justify the assertion that declining EBF coverage in urban settings, there is a need to provide statistical background of rate and past year trend of a) EBF rate, b) early skin-to-skin contact immediate after birth, and c) proper complementary feeding of children aged 6-23 months where data allows which breakdown by at least urban and rural.
2. The background should also provide nutritional status and trend among <5 year children comparing urban and rural settings. Nutritional status is a key reflection of adequate and quality complementary feeding. It will be helpful for the audience to understand this national dataset. The paper indicates that the LSIS II combines modules of MICS and DHS; how about LSIS I. What is the frequency of LSIS – when was LSIS I conducted and when the next LSIS will be conducted. LSIS III will be a combination of MICS and DHS or not etc. Also the author can provide comments on the national dataset of MICS and DHS separately, whether or not the combination of MICS and DHS should continue in Lao and should replicate to other LMIC
3. Discussion may need to provide reflection of government and partners' efforts which support, promote (such as baby friendly hospitals through ANC, intra- and post-partum) and protection breast feeding (notably maternity leave with pay among working mother in formal sector), against the backdrop of the effectiveness of implementing the Code of

Marketing BMS, and consistent Code violation by BMS industry. This discussion point is critical to understand the contextual environment surrounding the urban mothers in addition to quantitative evidence from Social Indicator survey.

Q 2 Please summarize the main findings of the study.

The findings confirm large disparities in breastfeeding practices, EBF for < 6 months and appropriate complementary feeding until 23 months among three types of residence where large-urban setting had much lower performance than small urban and rural areas.

Q 3 Please highlight the limitations and strengths.

The limitations are clearly identified and discussed thoroughly

PLEASE COMMENT

Q 4 Is the title appropriate, concise, attractive?

Title is clear and concise.

Q 5 Are the keywords appropriate?

Keywords are appropriate though need to add Lao PDR.

Q 6 Is the English language of sufficient quality?

No answer given.

Q 7 Is the quality of the figures and tables satisfactory?

Yes.

Q 8 Does the reference list cover the relevant literature adequately and in an unbiased manner?)

No answer given.

QUALITY ASSESSMENT

Q 9 Originality

Q 10 Rigor

Q 11 Significance to the field

Q 12 Interest to a general audience

Q 13 Quality of the writing

Q 14 overall scientific quality of the study

REVISION LEVEL

Q 15 Please take a decision based on your comments:

Minor revisions.